

# Low-dose hexavalent chromium induces mitophagy in rat liver via the AMPK-related PINK1/Parkin signaling pathway

Ningning Li[1,*], Xiaoying Li[1,*], Xiuzhi Zhang[1], Lixia Zhang[2], Hui Wu[3], Yue Yu[4], Guang Jia[5] and Shanfa Yu[6]

[1] Department of Pathology, Henan Medical College, Zhengzhou, Henan, China

[2] Department of Occupational Health and Environmental Health, College of Public Health, Zhengzhou University, Zhengzhou, Henan, China

[3] The Third People's Hospital of Henan Province, Zhengzhou, Henan, China

[4] National Institute for Occupational Health and Poison Control, Chinese Center for Disease Control and Prevention, Beijing, China

[5] Department of Occupational and Environmental Health Sciences, School of Public Health, Peking University, Beijing, China

[6] School of Public Health, Henan Medical College, Zhengzhou, Henan, China

[*] These authors contributed equally to this work.

Corresponding author
Shanfa Yu, shanfa_yu@126.com

## ABSTRACT

Hexavalent chromium (Cr(VI)) is a hazardous metallic compound commonly used in industrial processes. The liver, responsible for metabolism and detoxification, is the main target organ of Cr(VI). Toxicity experiments were performed to investigate the impacts of low-dose exposure to Cr(VI) on rat livers. It was revealed that exposure of 0.05 mg/kg potassium dichromate ($K_2Cr_2O_7$) and 0.25 mg/kg $K_2Cr_2O_7$ notably increased malondialdehyde (MDA) levels and the expressions of P-AMPK, P-ULK, PINK1, P-Parkin, and LC3II/LC3I, and significantly reduced SOD activity and P-mTOR and P62 expression levels in liver. Electron microscopy showed that CR(VI) exposure significantly increased mitophagy and the destruction of mitochondrial structure. This study simulates the respiratory exposure mode of CR(VI) workers through intratracheal instillation of CR(VI) in rats. It confirms that autophagy in hepatocytes is induced by low concentrations of CR(VI) and suggest that the liver damage caused by CR(VI) may be associated with the AMPK-related PINK/Parkin signaling pathway.

## INTRODUCTION

Chromium (Cr) is a hard, steel-gray metal that exists in various oxidation states ranging from Cr(II) to Cr(VI). Among these, Cr(III) and Cr(VI) are the most common and highly stable forms. Cr(VI) compounds are identified as one of the 17 chemicals posing a threat to humans by the United States Environmental Protection Agency (US EPA) (*McCullough et al., 1999*). Cr(VI) compounds are classified by the International Agency for Research on Cancer (IARC) as carcinogenic to humans (Group I) (*IARC, 2021*). Hexavalent chromium (Cr(VI)) is widely used across industries in welding, hot working stainless steel processing, chrome plating, spray painting, and coating activities (*Ndaw et al., 2022*). Cr(VI) enters

various environmental systems (air, water, and soil) through some natural processes and anthropogenic activities. It can lead to contamination and destruction of the ecosystem. Exposure to Cr(VI) causes severe effects on flora and fauna (*Jobby et al., 2018*; *Prasad et al., 2021*; *Pandey, Gautam & Singh, 2023*). Cr(VI) is responsible for multiorgan damage, such as kidney damage, liver failure, heart failure, skin disease, and lung dysfunction. The liver is considered one of the most important target organs for Cr(VI) toxicity (*Chakraborty et al., 2022*; *Singh et al., 2022*).

Cr(VI) can exert cellar toxicity through various mechanisms, and oxidative stress is one of the important pathways (*Renu et al., 2021*). The mitochondria generates energy in the form of ATP through the process of oxidative phosphorylation. Additionally, mitochondria are recognized as the main sources of reactive oxygen species (ROS), and actively participate in the regulation of cellular redox processes and ROS signaling (*Spinelli & Haigis, 2018*; *Kuznetsov et al., 2022*). Excess ROS can trigger mitochondrial dysfunction and mitochondrial autophagy (mitophagy). Mitophagy refers to the process in which aging or damaged mitochondria are specifically wrapped into the autophagosome under the stimulation of various harmful factors, and then degraded by lysosomes (*Sulkshane et al., 2021*). The vicious circle between mitochondrial dysfunction and oxidative stress is a key contributor to the progression of almost all hepatic damage (*Xu & Feng, 2023*).

There have been few studies on Cr(VI)-induced mitophagy, and its underlying molecular mechanism remains unclear. The PINK1/Parkin signaling pathway is an important mitophagy pathway that is mediated by PTEN-induced putative kinase1 (PINK1) and E3 ubiquitin ligase PARK2 (Parkin). When cells function normally, PINK1 is maintained at a low level due to mitochondrial import, protease cleavage, and proteasome degradation. When cells are damaged, PINK1 is not easily degraded but stably exists in the outer membrane of mitochondria, and recruits and activates Parkin from the cytoplasm. The activated Parkin covers the damaged mitochondria with ubiquitin, thus activating mitophagy (*Wang, Lu & Shen, 2020*; *Gan et al., 2022*; *Gladkova et al., 2018*).

It is possible that Cr(VI) participates in the process of liver injury through the regulation of mitophagy and oxidative stress in hepatocytes. However, the correlation between liver injury induced by Cr(VI), mitophagy, and their regulations remains unclear. Therefore, we used inhalable intratracheal instillation of Cr(VI) on rats in this study in order to explore the role of PINK1/Parkin pathway-mediated mitophagy in liver injury due to Cr(VI) exposure in rats.

Because of the promotion and implementation of occupational health and hygienic environment standards, there typically is low Cr(VI) exposure in industrial and living environments. In this study, we explored the toxic effects of low concentration and short time Cr exposure on rat liver.

## MATERIAL AND METHODS

### Animals and treatment
Four-week-old adult male Sprague-Dawley (SD) rats (280 ± 50 g) were supplied by Jinan Pangyue Laboratory Animal Breeding Co. Ltd (Jinan, China). They were housed at a

controlled temperature (22.0 ± 1.0 °C) with 12 h light/dark cycle and provided with standard food and water. After acclimation for one week, rats were randomly divided into three groups, with four rats in each group. Sample size was selected according to the 3R principle (*Sneddon, Halsey & Bury, 2017*). All procedures were conducted in accordance with the Guidelines for the Care and Use of Laboratory Animals published by the Ministry of Health of People's Republic of China. All experimental protocols were approved by the Ethics Committee of Zhengzhou University (Ethics Review Number: ZZUIRB:2022-55).

## Experimental protocol

The most common route for Cr(VI) occupational exposure is inhalation (*Leese et al., 2023*). As it is simple, quick, and allows control of the applied dose, intratracheal instillation has been employed in many studies as an alternative exposure procedure. In this study, we adopted the method of intratracheal instillation poisoning (*Gutierrez et al., 2023*; *Driscoll et al., 2000*). Twelve healthy rats were randomly assigned into three groups, with four rats in each group: the saline group, the 0.05 mg/kg Cr(VI) group, and the 0.25 mg/kg Cr(VI) group. The dosages applied here were determined from time weighted average (TWA) and inhalation-to-drip dose conversion formula (*Song et al., 2014*). Random number generation was performed using R 3.6.2 software (*R Core Team, 2020*). The intratracheal instillation method and experimental process were referenced from a previous study (*Zhang et al., 2023*). Rats received weekly inhalable intratracheal instillation of potassium dichromate ($K_2Cr_2O_7$) dissolved in sterile 0.9% sodium chloride solution at 0, 0.05, or 0.25 mg Cr/kg body weight for 28 days (a total of five times). The rats were fasted and abstained from water 24 h before sacrifice. The rats were deeply anesthetized *via* intraperitoneal injection of 1% sodium pentobarbital (70 mg/kg) 24 h after the final instillation. Blood samples were collected from the abdominal aorta of the rats. The rats were euthanized by means of cervical dislocation, and the death of the animals was confirmed by testing for loss of pain response. Rat livers were harvested immediately. Testing and data analysis were performed by different staff members and group allocation was blind for them.

## Liver function

Liver function tests can help determine hepatic injury and liver disease diagnosis. Here, the entire blood specimen was applied to detect liver function of the rats. The experimental processes were described in a previous study (*Li et al., 2024*).

## Liver tissue ultrastructure examination

Electron microscopy is one of the best approaches that can directly provide the ultrastructure evidence for mitophagy. The liver tissues were processed into electron microscope sections and observed using an electron microscope (TEM) (*Li et al., 2024*).

## Liver tissue oxidative stress assay

Oxidative stress is the initial stage of liver injury, and super-oxide dismutase (SOD) and malondialdehyde (MDA) can indicate the oxidative stress intensity. After standing for 30 min, the serum was collected by centrifugation at 3,000 rpm for 10 min. Samples were stored at −20 °C and measured as soon as possible. The levels of SOD and MDA in the

serum were determined according to the instructions of a SOD assay kit (A001-3; Nanjing Jiancheng Institute of Biological Engineering, China), and MDA assay kit (A003-1; Nanjing Jiancheng Institute of Biological Engineering, China).

### Analysis of relevant protein levels by Western blotting (WB)

The PINK1/Parkin pathway proteins and marker proteins of autophagy were detected using WB. The WB analysis methods were described in a previous study (*Li et al., 2024*). WB kits were purchased from Jiangsu Kegel Biotechnology Co., Ltd., China. The following antibodies were used: Mouse Anti-PINK1 (ab186303; Abcam, Cambridge, UK), Rabbit Anti-P-Parkin (PA5-114616; Thermo Fisher Scientific, Waltham, MA, USA), Rabbit Anti-P-AMPK (AF5908; Beyotime, Jiagsu, China), Mouse Anti-P-mTOR (67778-1-Ig; Proteintech, Rosemont, IL, USA), Rabbit Anti-P-ULK1 (ab203207; Abcam, Cambridge, UK), Rabbit Anti-LC3I (18722-1-AP; Proteintech, Rosemont, IL, USA), Rabbit Anti-LC3II (81004-1-RR; Proteintech, Rosemont, IL, USA), Rabbit Anti-P62 (ab91526; Abcam, Cambridge, UK), Rabbit Anti-Beclin (ab207612; Abcam, Cambridge, UK), and Rabbit Anti-GAPDH (KGAA002; KeyGEN, Seattle, WA, USA). The target proteins were developed using enhanced chemiluminescence (ECL) reagents. Imaging was performed using the ChemiDoc MP Imaging System and the results were analyzed in grayscale using Gel-Pro32 software.

### Statistical analysis

All statistical analyses were performed using SPSS 25.0 (IBM, Armonk, NY, USA). If normality and equal variance passed, differences among groups were analyzed using one-way analysis of variance (ANOVA). Nonparametric data were analyzed by the Kruskal-Wallis ANOVA-based test on ranks followed by Dunn's post-hoc test. A difference was considered significant when $P < 0.05$ and highly significant when $P < 0.01$. Data were graphed with GraphPad PRISM 8.0 software, and bar graphs generated from this analysis demonstrate means $\pm$ SD.

## RESULTS

### Biochemical indices after Cr(VI) exposure

Figure 1 shows the changes of biochemical indices in the serum after rats were exposed to Cr(VI). Compared with the saline group, aspartate aminotransferase/ alanine aminotransferase (AST/ALT), albumin (ALB), and total biliary acid (TBA) levels exhibited no significant difference.

### Oxidative stress indices in liver tissue

In the 0.05 mg/kg Cr(VI) group and 0.25 mg/kg Cr(VI) group, SOD activity was decreased. MDA increased in the 0.25 mg/kg Cr(VI) group (Fig. 2). Results suggested that the continuous accumulation of Cr(VI) in livers induced liver cell oxidative stress.

### TEM images of liver tissues

In the saline group, mitochondria showed good morphology and mitochondrial cristae were arranged neatly. In the 0.05 mg/kg Cr(VI) group and 0.25 mg/kg Cr(VI) group,

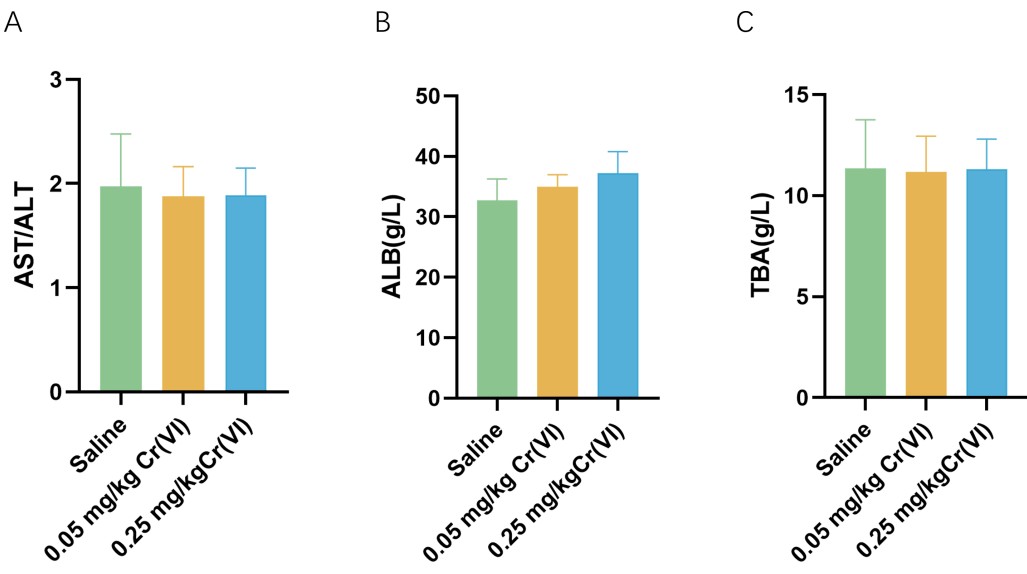

**Figure 1  Serum biochemical indices of liver function in rats.** (A–C) Detection of serum AST/ALT, ALB, and TBA levels in rats. Data are expressed as the M ± SD ($n = 4$).

some mitochondrial cristae were ruptured and disordered (red arrows indicate disrupted mitochondrial cristae) and the number of autophagosomes increased in a dose-dependent manner (yellow arrows indicate autophagosomes) (Fig. 3).

### PINK1/Parkin pathway expression

To further investigate the level of PINK/Parkin pathway after treatment with Cr(VI), we measured the levels of PINK and P-Parkin using WB. Figure 4 shows that compared with those of the saline group, the protein levels of PINK1 and P-Parkin of the 0.05 mg/kg Cr(VI) and 0.25 mg/kg Cr(VI) groups were significantly increased.

### AMPK/mTOR/ULK1 pathway expression

We measured the levels of related proteins AMP-activated protein kinase (P-AMPK), P-mTOR (rapamycin), and UNC-51-like kinase 1 (P- ULK1) using WB. Figure 5 shows that the protein levels of P-AMPK and P-ULK1 in the 0.05 mg/kg Cr(VI) and 0.25 mg/kg Cr(VI) groups were significantly increased compared with those of the saline group. The expression of P-mTOR was significantly lower in the 0.05 mg/kg Cr(VI) and 0.25 mg/kg Cr(VI) groups than that in the saline group.

### Autophagy protein expression

WB shows that, compared with those in the saline group, the levels of the autophagy marker microtubule-associated protein 1 light chain 3 (LC3)II/LC3I and the early autophagy proteins Beclin1 were significantly higher in the Cr(VI) exposure groups. Cr(VI) exposure significantly decreased P62 level compared with those in the saline group. The expression of P62 was significantly lower in the 0.05 mg/kg Cr(VI) group than that in the 0.25 mg/kg Cr(VI) group (Fig. 6).

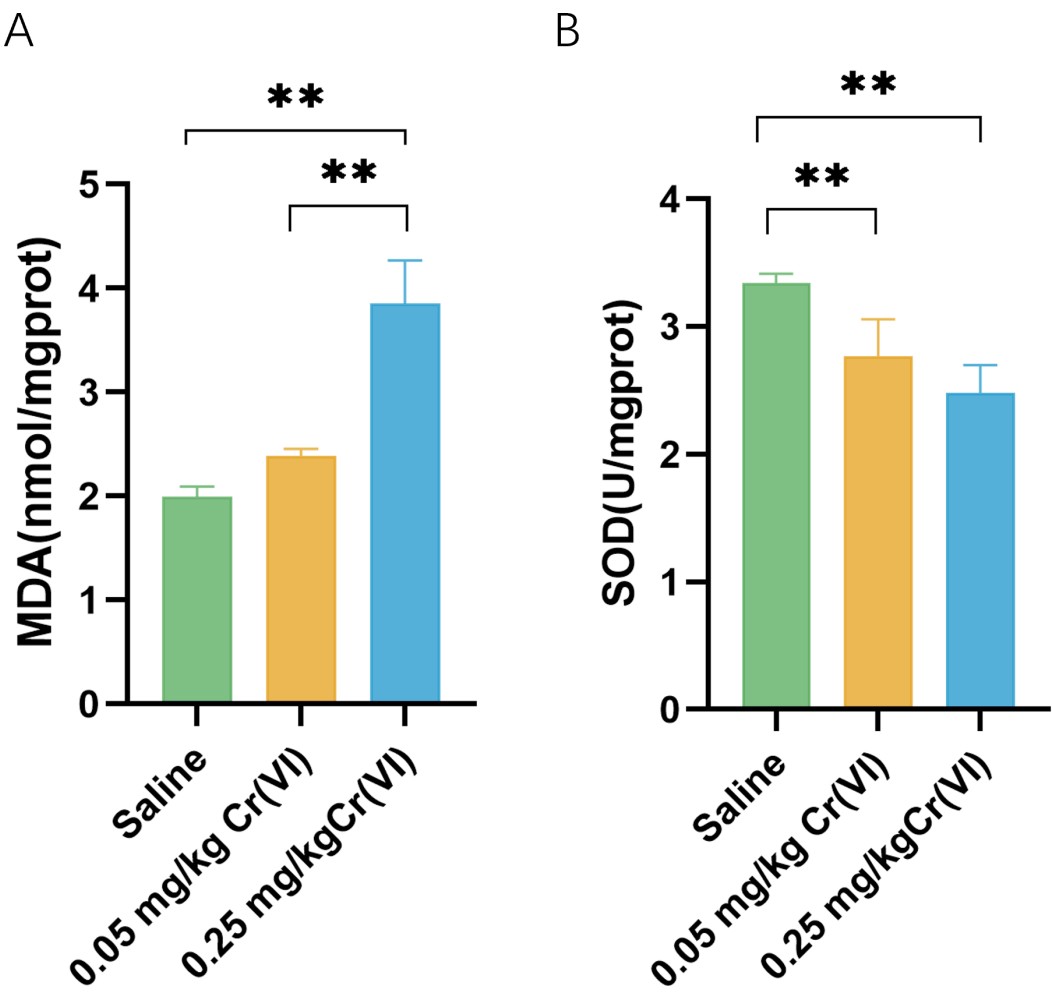

**Figure 2** **Intracellular MDA contents and SOD activity of liver tissues.** (A) MDA contents of liver tissues. (B) SOD activity of liver tissues. MDA contents and SOD activity are expressed as the M ± SD ($n = 4$). Statistical significance is indicated by *, $P < 0.05$, **, $P < 0.01$.

## DISCUSSION

Cr(VI) is the most common and toxic valence state of Cr. Many studies have shown that the liver is the main target organ for Cr(VI) toxicity (*Anandasadagopan et al., 2017*; *Xiao et al., 2018*). This study revealed that Cr(VI) exposure notably increased the MDA level and expressions of P-AMPK, P-ULK, PINK1, P-Parkin, and LC3II/LC3I, while significantly reduced SOD activity and P-mTOR and P62 expression levels. The results of electron microscopy showed that Cr(VI) exposure caused a significant increase in mitophagy and the destruction of mitochondrial structure. In brief, this study found that exposure to Cr(VI) could lead to liver oxidative stress and mitophagy.

Cr(VI) is water-soluble and easily penetrates the cell membrane into the cell. Under the action of reducing agents such as glutathione in the cell, Cr(VI) is reduced to Cr(III). At the same time, oxygen free radical ROS is induced to cause cell lipid peroxidation, which is

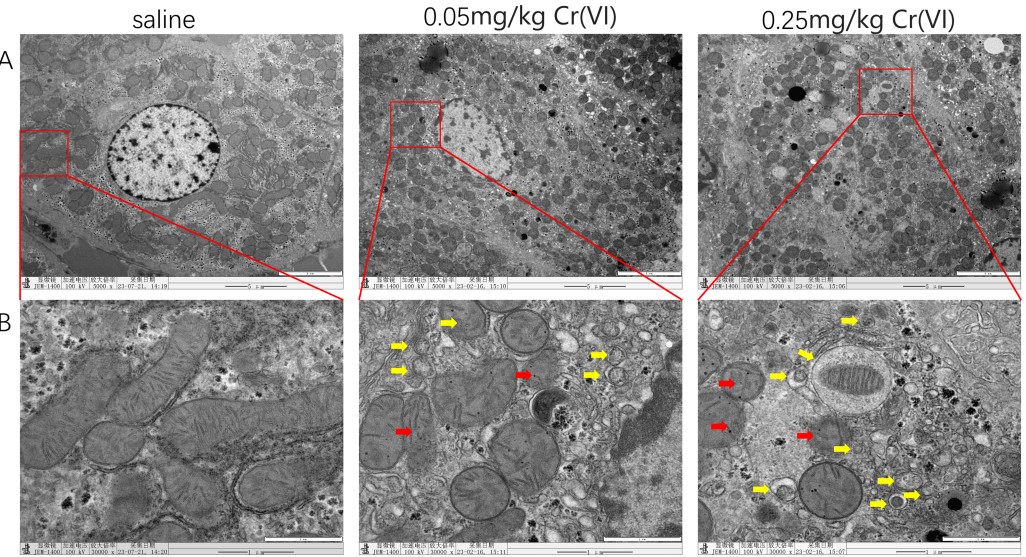

**Figure 3** **TEM images of liver tissues.** (A) Ultrastructure changes of nucleus (scale bars, 5 μm). (B) Ultrastructure changes of the mitochondria and autophagosomes (scale bars, 1 μm). Red arrows indicate disrupted mitochondrial cristae and yellow arrows indicate autophagosomes.

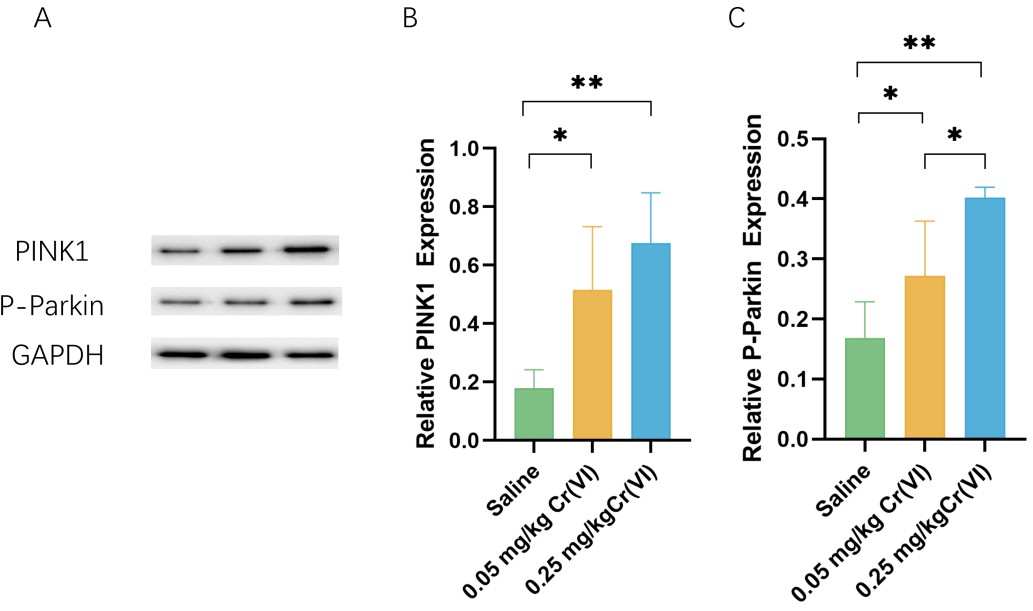

**Figure 4** **Western blot analysis of the expression of PINK1 and P-Parkin proteins in rat liver tissues.** (A) A representative immunoblotting of PINK1 and P-Parkin. GADPH was used as an internal reference for proteins. (B–C) The ratio of PINK1 and P-Parkin respectively. Protein levels are expressed as the M ± SD ($n = 4$). Statistical significance is indicated by *, $P < 0.05$, **, $P < 0.01$.

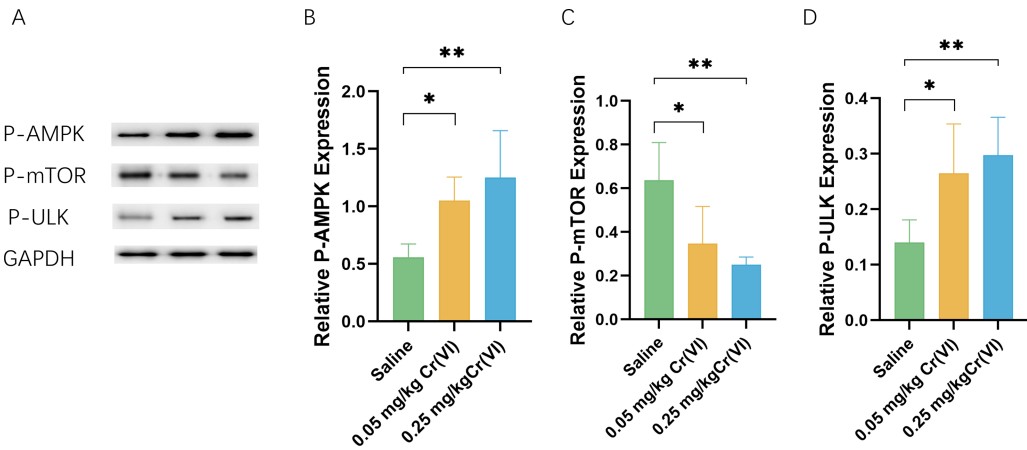

**Figure 5** (A) A representative immunoblotting of P-AMPK, P-mTOR, and P-ULK1. GADPH was used as an internal reference for proteins. (B–D) The ratio of P-AMPK, P-mTOR, and P-ULK1 respectively. Protein levels are expressed as the M ± SD ($n = 4$). Statistical significance is indicated by *, $P < 0.05$, **, $P < 0.01$.

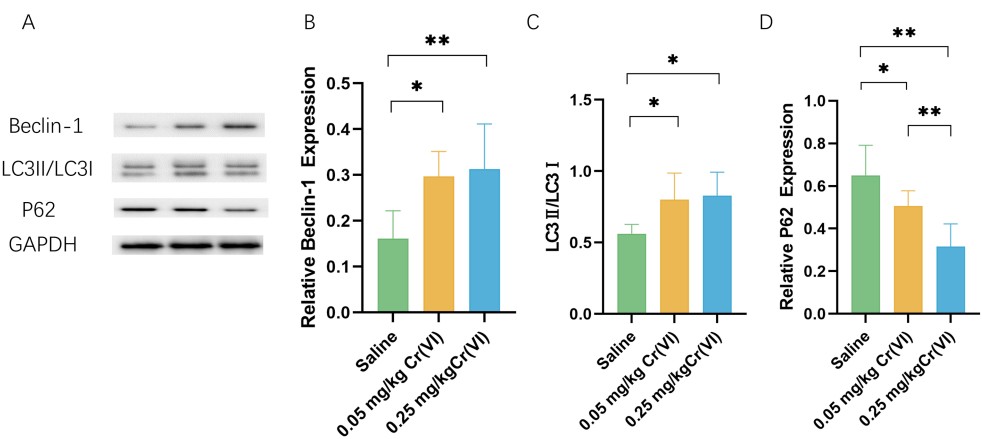

**Figure 6** Western blot analysis of the expression of autophagy-related proteins in rat liver tissues. (A) A representative immunoblotting of Beclin-1, LC3II/LC3I, and P62. GADPH was used as an internal reference for proteins. (B–D) The ratio of Beclin-1, LC3II/LC3I, and P62 respectively. Protein levels are expressed as the M ± SD ($n = 4$). Statistical significance is indicated by *, $P < 0.05$, **, $P < 0.01$.

one of the most important reasons for Cr(VI) to cause liver damage. MDA is the product of hepatic lipid peroxidation and reflects the intensity of hepatic oxidative stress. In this experiment, MDA in 0.25 mg/kg Cr(VI) group was significantly higher than that in saline group ($P < 0.05$), suggesting that after Cr(VI) enters the liver and causes lipid peroxidation, resulting in product accumulation. To reduce oxidative stress, cells activate antioxidant enzymes to protect against this oxidative damage. SOD is an important antioxidant enzyme in the body. In this study, SOD activity in 0.05 mg/kg Cr(VI) and 0.25 mg/kg Cr(VI) groups

decreased significantly ($P < 0.05$), suggesting that oxidative stress occurred in the liver, resulting in a decrease in antioxidant enzyme activity.

Under the stimulation of oxidative stress and other factors, the mitochondria in the cell undergo depolarization damage. The damaged mitochondria are wrapped into the autophagosome by specific membranes mediated by autophagy proteins and bind to lysosomes, thus completing mitochondrial degradation. Mitophagy is an evolutionarily conserved cellular process in which cells maintain energy metabolic balance by removing dysfunctional or redundant mitochondria (*Palikaras, Lionaki & Tavernarakis, 2018*) and contribute to mitochondrial quality improvement (*Pickles, Vigié & Youle, 2018*; *Kiriyama & Nochi, 2017*).

Mitochondria are the main source of ROS in cells, and the mitochondria itself is a very sensitive target of ROS (*Kuznetsov et al., 2022*). Mitophagy may function more broadly to limit the deleterious effects of ROS on cellular functions (*Ma et al., 2020*). However, other data suggest that the process of mitophagy may, in some cases, increase mitochondrial ROS (mtROS) levels that might trigger a cell to further induce mitophagy and therefore propagate the elevation in mtROS levels through a positive feedback loop (*Schofield & Schafer, 2021*). The vicious circle between mitochondrial dysfunction and oxidative stress is a key contributor to liver injury.

The PINK1/Parkin pathway is the important mechanism of mitophagy (*Killackey, Philpott & Girardin, 2020*). When the body is stimulated by factors such as toxicity and aging, the structure and function of mitochondria are disturbed, resulting in oxidative stress. Excessive production of ROS can activate PINK1 kinase activity (*Xiao et al., 2018*; *Xiao et al., 2017a*; *Xiao et al., 2017b*). PINK1 then phosphorylates and activates Parkin's ubiquitin E3 ligase activity, which labels damaged mitochondria with ubiquitin prior to mitophagy (*McWilliams & Muqit, 2017*; *Pickles, Vigié & Youle, 2018*). ROS induces mitophagy by activating the PINK1/Parkin pathway and removing excess mitochondria to maintain the healthy operation of the mitochondrial system. Mitophagy is essential for mitochondrial redox balance, mitochondrial function, and cell homeostasis. In this study, PINK1 protein in the Cr(VI) exposed groups was significantly higher than in the saline group ($P < 0.05$), which proved that liver tissue was damaged after Cr exposure, causing the retention of undegraded PINK1 protein in the mitochondrial outer membrane. At the same time, P-Parkin is an activated Parkin protein, and P-Parkin is significantly increased in the Cr(VI) exposed groups ($P < 0.05$), also confirming that autophagy was initiated by hepatocytes through the PINK1/Parkin pathway.

AMPK is a key regulator of PINK1/Parkin dependent mitophagy. *Egan et al. (2011)* first revealed that AMPK is involved in mitophagy. AMPK is a highly conserved cellular energy volume and nutrient status sensor in eukaryotic cells. AMPK monitors the ratio of AMP/ATP (or ADP/ATP) and acts to restore energy homeostasis by switching on alternate catabolic pathways that produce ATP, while shutting down biosynthetic pathways and other non-essential processes that consume ATP (*Sharma et al., 2023*; *Hardie, 2014*). mTOR is a mechanistic target at the intersection of synthesis and catabolism, promoting cell growth by stimulating the biosynthetic pathway and inhibiting catabolism by reducing autophagy. Its close signaling interplay with the energy sensor AMPK dictates whether the
cell favors anabolic or catabolic processes to maintain cellular homoeostasis (*Rabanal-Ruiz, Otten & Korolchuk, 2017*). The AMPK-mTOR pathway is an important signaling pathway in autophagy because AMPK suppresses mTOR activity and mTOR inhibits ULK1. Additionally, AMPK directly phosphorylates ULK1 to trigger autophagy. AMPK controls ULK1 *via* a two-pronged mechanism (*Tamargo-Gómez & Mariño, 2018*; *Zhang & Lin, 2016*). The results of this experiment confirm that Cr(VI) exposure activates this pathway.

As a master sensor of cellular stress, AMPK is activated and downstream substrate ULK1 are phosphorylated, which leads to specific Parkin phosphorylation to activate mitophagy (*Guo et al., 2023*). AMPK can activate both PINK1 kinase and Parkin E3 ligase through phosphorylation (*Wang et al., 2018*; *Lee et al., 2019*). In short, there are complex interactions between PINK1/Parkin pathways and AMPK/mTOR/ULK1 pathway that affect mitophagy.

Autophagy is a highly dynamic, multi-step biological process. LC3 is divided into LC3II and LC3I, which is a key protein in the formation of mammalian autophagosomes. When autophagy occurs, LC3I binds to phosphatidyl ethanolamine and transforms into LC3II, which is located in the outer membrane of autophagy, so LC3II is generally considered to be the signature protein of autophagy (*Martinez et al., 2015*). The increase of LC3II/LC3I ratio indicates the activation of autophagy flow and is also an important indicator for evaluating autophagy level (*Gong et al., 2020*). The results of this study showed that compared with the saline group, the LC3II/LC3I ratio was significantly increased in both dose groups ($P < 0.05$), suggesting that liver damage caused by Cr(VI) exposure may promote the activation of autophagy. Beclin1 is an essential molecule in the formation of autophagosomes, which can mediate the localization of other autophagy proteins to autophagosomes, thereby regulating the formation and maturation of autophagosomes (*Xu & Qin, 2019*). In our experimental study, Cr(VI) toxicity caused a significant increase in Beclin1 protein ($P < 0.05$), also confirming that the activation of autophagy is caused by exposure. Ubiquitin-binding protein P62 decrease in the Cr(VI) exposed groups in our study was another proof of autophagy. Transmission electron microscopy (TEM) observation is a powerful method used to verify mitophagy (*Chakraborty et al., 2020*). The results of TEM observation in this experiment showed that mitophagy increased significantly in the two Cr(VI) exposed groups, suggesting that Cr(VI) exposure can induce mitochondrial disruption and mitophagy in hepatocytes.

Cr is a toxic heavy metal. In recent years, the few studies on mitophagy caused by Cr(VI) have been *in vitro* (*Zhang et al., 2020*; *Xu et al., 2020*). Similar to our findings, these *in vitro* experiment findings indicate that Cr(VI) may contribute to mitochondrial morphology and function damage and may therefore lead to mitophagy. Mitophagy is closely related to tissue damage, immune response, aging, and tumor growth inhibition (*Onishi et al., 2021*). Mitophagy is the stress and self-rescue of liver cells after Cr(VI) infection, and even further apoptosis and necrosis, resulting in more serious liver damage and even liver diseases (*Ke, 2020*). In fact, occupational exposure to Cr(VI) does pose a health hazard (*Hessel et al., 2021*; *den Braver-Sewradj et al., 2021*). There are many regulatory pathways for mitophagy, and it is of great significance to determine the role of proteins in these pathways to find suitable marker proteins for Cr exposure effects and suitable molecular targets for health

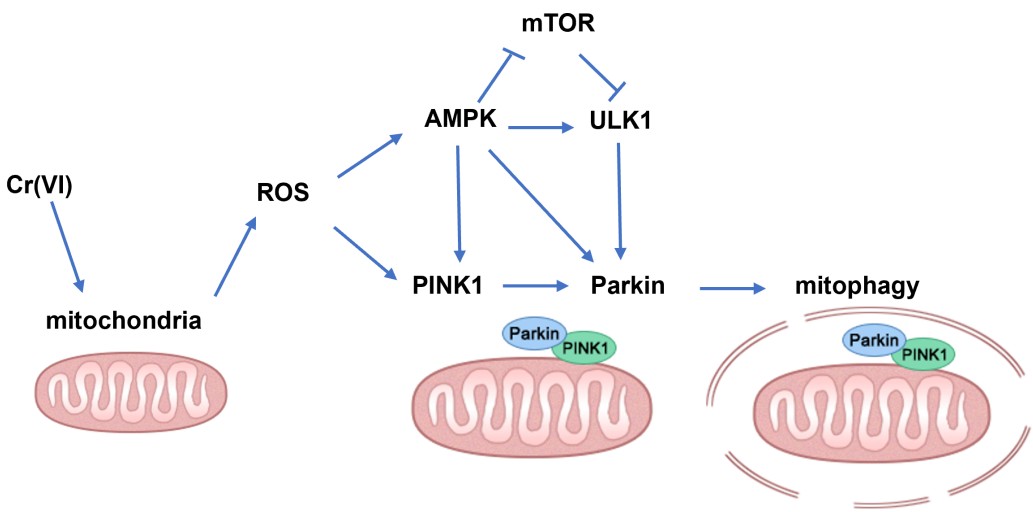

**Figure 7** **Schematic diagram of the molecular mechanism.** An arrow ($\rightarrow$) indicates possible promoting effect; $\perp$ indicates possible inhibiting effect.

intervention of Cr-exposed workers. This study shows the poisoning of Cr(VI) workers *via* inhalable intratracheal instillation of Cr(VI), that the autophagy of hepatocytes is caused by low concentration Cr(VI) poisoning, and that liver damage caused by Cr(VI) may be related to the AMPK-related PINK/Parkin signaling pathway. It provides experimental basis and theoretical basis for liver injury caused by Cr pollution and health monitoring of workers.

However, our study had several limitations. The exposure time was not very long, and further studies are needed to explore the correlation between liver injury and PINK/Parkin signaling pathway proteins and autophagy by extending the exposure time according to the results of this experiment. As a result, further experiments and more in-depth investigations are required to find the effect markers of PINK/Parkin pathway in liver injury, and their application to the health monitoring of workers with occupational exposure to Cr(VI) will be our future work direction.

## CONCLUSION

In summary, our findings demonstrate that low concentration Cr(VI) could activate the PINK1/Parkin pathway and up-regulate mitophagy in rat liver (the possible mechanisms are shown in Fig. 7). Therefore, we concluded that the PINK1/Parkin-pathway *via* AMPK might play an important role in Cr-induced liver injury and can be used as a potential target for the treatment of Cr-induced liver injury.

### Funding

This work was supported by the National Natural Science Foundation of China (No. U2004202). The funders had no role in study design, data collection and analysis, decision to publish, or preparation of the manuscript.

### Grant Disclosures

The following grant information was disclosed by the authors:
The National Natural Science Foundation of China: No. U2004202.

### Competing Interests

The authors declare there are no competing interests.

### Author Contributions

- Ningning Li conceived and designed the experiments, performed the experiments, analyzed the data, prepared figures and/or tables, and approved the final draft.
- Xiaoying Li conceived and designed the experiments, performed the experiments, analyzed the data, prepared figures and/or tables, and approved the final draft.
- Xiuzhi Zhang analyzed the data, prepared figures and/or tables, and approved the final draft.
- Lixia Zhang performed the experiments, prepared figures and/or tables, and approved the final draft.
- Hui Wu performed the experiments, prepared figures and/or tables, and approved the final draft.
- Yue Yu performed the experiments, authored or reviewed drafts of the article, and approved the final draft.
- Guang Jia conceived and designed the experiments, authored or reviewed drafts of the article, and approved the final draft.
- Shanfa Yu conceived and designed the experiments, authored or reviewed drafts of the article, and approved the final draft.

### Animal Ethics

The following information was supplied relating to ethical approvals (i.e., approving body and any reference numbers):
Ethics Committee of Zhengzhou University.

### Data Availability

WB electrophoresis bands are available in the Supplementary File.

## Supplemental Information

Supplemental information for this article can be found online at http://dx.doi.org/10.7717/peerj.17837#supplemental-information.

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

# PeerJ

**Schofield JH, Schafer ZT. 2021.** Mitochondrial reactive oxygen species and mitophagy: a complex and nuanced relationship. *Antioxidants & Redox Signal* **34(7)**:517–530 DOI 10.1089/ars.2020.8058.

**Sharma A, An SK, Singh N, Dwivedi UN, Kakkar P. 2023.** AMP-activated protein kinase: an energy sensor and survival mechanism in the reinstatement of metabolic home-ostasis. *Experimental Cell Research* **428(1)**:113614 DOI 10.1016/j.yexcr.2023.113614.

**Singh V, Singh N, Verma M, Kamal R, Tiwari R, Sanjay Chivate M, Rai SN, Kumar A, Singh A, Singh MP, Vamanu E, Mishra V. 2022.** Hexavalent-chromium-induced oxidative stress and the protective role of antioxidants against cellular toxicity. *Antioxidants* **11(12)**:2375 DOI 10.3390/antiox11122375.

**Sneddon LU, Halsey LG, Bury NR. 2017.** Considering aspects of the 3Rs principles within experimental animal biology. *Journal of Experimental Biology* **220(Pt 17)**:3007–3016 DOI 10.1242/jeb.147058.

**Song Y, Wang T, Pu J, Guo J, Chen Z, Wang Y, Jia G. 2014.** Multi-element distribution profile in Sprague-Dawley rats: effects of intratracheal instillation of Cr(VI) and Zn intervention. *Toxicology Letters* **226(2)**:198–205 DOI 10.1016/j.toxlet.2014.02.008.

**Spinelli JB, Haigis MC. 2018.** The multifaceted contributions of mitochondria to cellular metabolism. *Nature Cell Biology* **20(7)**:745–754 DOI 10.1038/s41556-018-0124-1.

**Sulkshane P, Ram J, Thakur A, Reis N, Kleifeld O, Glickman MH. 2021.** Ubiquitination and receptor-mediated mitophagy converge to eliminate oxidation-damaged mito-chondria during hypoxia. *Redox Biology* **45**:102047 DOI 10.1016/j.redox.2021.102047.

**Tamargo-Gómez I, Mariño G. 2018.** AMPK: regulation of metabolic dynamics in the context of autophagy. *International Journal of Molecular Sciences* **19(12)**:3812 DOI 10.3390/ijms19123812.

**Wang B, Nie J, Wu L, Hu Y, Wen Z, Dong L, Zou MH, Chen C, Wang DW. 2018.** AMPK $\alpha$2 protects against the development of heart failure by enhancing mi-tophagy via PINK1 Phosphorylation. *Circulation Research* **122(5)**:712–729 DOI 10.1161/CIRCRESAHA.117.312317.

**Wang L, Lu G, Shen HM. 2020.** The long and the short of PTEN in the regulation of mi-tophagy. *Frontiers in Cell and Developmental Biology* **8**:299 DOI 10.3389/fcell.2020.00299.

**Xiao B, Deng X, Lim GGY, Xie S, Zhou ZD, Lim KL, Tan EK. 2017a.** Superox-ide drives progression of Parkin/PINK1-dependent mitophagy following translocation of Parkin to mitochondria. *Cell Death & Disease* **8(10)**:e3097 DOI 10.1038/cddis.2017.463.

**Xiao B, Goh JY, Xiao L, Xian H, Lim KL, Liou YC. 2017b.** Reactive oxygen species trigger Parkin/PINK1 pathway-dependent mitophagy by inducing mitochondrial recruitment of Parkin. *Journal of Biological Chemistry* **292(40)**:16697–16708 DOI 10.1074/jbc.M117.787739.

**Xiao Y, Zeng M, Yin L, Li N, Xiao F. 2018.** Clusterin increases mitochondrial respiratory chain complex I activity and protects against hexavalent chromium-induced cytotoxicity in L-02 hepatocytes. *Toxicology Research* **8(1)**:15–24.

**Xu HD, Qin ZH. 2019.** Beclin 1, Bcl-2 and Autophagy. *Advances in Experimental Medicine and Biology* **1206**:109–126 DOI 10.1007/978-981-15-0602-4_5.

**Xu J, Feng Z. 2023.** Role of oxidative stress in mitochondrial function: relevance for liver function. *Antioxidants* **12(9)**:1784 DOI 10.3390/antiox12091784.

**Xu Y, Wang X, Geng N, Zhu Y, Zhang S, Liu Y, Liu J. 2020.** Mitophagy is involved in chromium (VI)-induced mitochondria damage in DF-1 cells. *Ecotoxicology and Environmental Safety* **194**:110414 DOI 10.1016/j.ecoenv.2020.110414.

**Zhang CS, Lin SC. 2016.** AMPK promotes autophagy by facilitating mitochondrial fission. *Cell Metabolism* **23(3)**:399–401 DOI 10.1016/j.cmet.2016.02.017.

**Zhang L, Li N, Zhang X, Wu H, Yu S. 2023.** Hexavalent chromium caused DNA damage repair and apoptosis via the PI3K/AKT/FOXO1 pathway triggered by oxidative stress in the lung of rat. *Ecotoxicology and Environmental Safety* **267**:115622 DOI 10.1016/j.ecoenv.2023.115622.

**Zhang Y, Bian H, Ma Y, Xiao Y, Xiao F. 2020.** Cr(VI)-induced overactive mitophagy contributes to mitochondrial loss and cytotoxicity in L02 hepatocytes. *Biochemical Journal* **477(14)**:2607–2619 DOI 10.1042/BCJ20200262.