# Peer review of "Low-dose hexavalent chromium induces mitophagy in rat liver via the AMPK-related PINK1/Parkin signaling pathway"

_PeerJ, doi:10.7717/peerj.17837_

## Round 0.1 · original submission · Major Revisions

Having been appraised by three reviewers, we have determined a major revision is needed. In particular, reviewer 2 has provided a comprehensive evaluation - we look forward to your response in due course.

Reviewer 1 ·

Basic reporting

The language is clear, unambiguous and professional throughout. One typographical error in line 106 "precious" should be "previous".
The literature references provide sufficient background and context in the field.
The article, figures and tables are structured professionally.
The results are self-contained with relevant results to the hypotheses. It seems to be a follow on of work based on very similar treatments - reference is Li et al 2023 used in the reference list.
On line 67, the authors mention a process of "mitochondrial endocytosis". I am not sure what is meant by this phrase, can the authors provide a reference for this, and the other mechanisms of PINK1 level maintenance in normal cells.

Experimental design

The article provides original primary research that is a continuation of the work of the group into the mechanism of Chromium toxicity in the liver.
The research fills a knowledge gap by using the rat as a model system to study chromium toxicity in the liver.
The study is based on 3 treatments each containing 4 animals based on 3R recommendations. However, these is not an indication of whether this was determined using power analysus.
Enzyme activity assays and western blot quantification are based on 4 repeats, presumably from the 4 individual animals used in the study.
The methods are described sufficiently to be replicated.

Validity of the findings

The results display a modest change in the measured parameters. There is very little change to liver function markers and there is at most a 3 fold change in any of the samples measured by western blotting.
The data has been provided and are controlled.
The conclusion is based on some information not presented in the paper, but from previous literature. The concluding figure 7 has a complex network of protein interactions. The protein levels were measured but there is no cause and effect to demonstrate the protein network is actually taking place in this model.
To demonstrate how the impact of phosphorylation of target proteins seen in Figures 4 and 5, the total protein amounts (for Parkin in Figure 4; and for AMPK, mTOR and ULK in Figure 5) should be shown and the ratio of phosphorylated to non-phosphorylated should also be shown, as this could demonstrate a difference that can highlight that phosphorylation cascade is altered.

Reviewer 2 ·

Basic reporting

no comment

Experimental design

no comment

Validity of the findings

no comment

Additional comments

This manuscript investigated the effects of low concentrations of hexavalent chromium on the liver of rats and explored its relationship with mitochondrial autophagy and the AMPK-related PINK1/Parkin signaling pathway. The study found that low concentrations of hexavalent chromium can activate the PINK1/Parkin signaling pathway, upregulate mitochondrial autophagy, and potentially be involved in chromium-induced liver damage through the AMPK signaling pathway. However, some weaknesses need to be improved. I believe that consideration of the following issues will improve the manuscript.
Comment 1: Prior to assessing the liver function parameters, did the authors have the experimental animals undergo fasting?
Comment 2: In the present manuscript, the authors employed intratracheal instillation of hexavalent chromium [Cr(VI)] to conduct the experiments. Given this route of exposure, the primary target organ for Cr(VI)-induced injury should be the lung. However, the focus of this study appears to be on the hepatotoxic effects of Cr(VI) exposure. The rationale for assessing liver outcomes, rather than pulmonary effects, following this mode of Cr(VI) administration should be explained in the Introduction.
Comment 3: In the 96th line, the author states that "The most common route of Cr(VI) exposure is inhalation." However, the referenced source does not support this claim, as it only addresses occupational exposure. To my knowledge, other common exposure routes for hexavalent chromium also include contact with contaminated environments and ingestion of contaminated foods, in addition to occupational exposure.
Comment 4: The authors did not describe the duration of the experiment in the Materials and Methods section. This information is crucial for ensuring the reproducibility of the study. The authors should provide more details about the experiment, including its duration, in the Materials and Methods section.
Comment 5: ALT and AST are enzymes that are normally predominantly contained within liver cells. If the liver is injured or damaged, the liver cells spill these enzymes into the blood, raising the AST and ALT enzyme levels in the blood. In this manuscript, the authors observed significant mitochondrial autophagy in rats exposed to hexavalent chromium, despite the absence of significant changes in AST/ALT levels. The authors need to provide more explanation for this discrepancy. Additionally, it is recommended that the authors report AST and ALT values separately, rather than using the ratio.
Comment 6: Please, check all the reference formats throughout the manuscript. Consult the journal's reference style for the exact appearance of these elements and the use of capitalization.
Comment 7: The manuscript has a few instances where punctuation has been omitted. For example, there is a missing period at the end of the sentence on line 116.
Comment 8: In the Materials and Methods section, the authors should provide the source information for all antibodies used in the study.
Comment 9: In the Discussion section, it is important to compare and contrast the results of this study with previous research to emphasize the novelty and significance of the current study.
Comment 10: During the experimental process, did the authors focus on the histopathological changes in the liver tissue of the experimental animals? Did they use common experimental techniques such as Hematoxylin and Eosin (HE) staining and Oil Red O staining to observe the damage to rat liver tissue caused by hexavalent chromium?

Reviewer 3 ·

Basic reporting

The experimental design of this study is scientific and reasonable, which can well elucidate the mechanism of cadmium-induced hepatotoxicity. I suggest it be published after minor revisions.The modification suggestions are as follows:
1. Abstract: Please add some background information about the study to highlight its significance.
2. Manuscript format: Please carefully check the format of the manuscript to ensure that it meets the requirements of the journal.
3. Figures: The color matching of the figure is not reasonable, it is recommended to modify.
4. The English language needs to be improved by a fluent English speaker.

Experimental design

The experimental design of this study is scientific and reasonable.

Validity of the findings

The experimental results were well verified.

Additional comments

.

---

## Round 0.2 · Minor Revisions

Your article is improved with your latest round of revisions. However, there are still significant issues with regards to the standard of writing and English grammar. Accordingly, we request you to engage with relevant native English language services or colleagues to help improve the standard of scientific writing.

Reviewer 1 ·

Basic reporting

Basic reporting is appropriate. There are still issues with English language and further proof-reading is required.

Experimental design

All aspectes of Experimental Design are appropriate.

Validity of the findings

The validity of findings is suitable.

---

## Round 0.3 · accepted · Accept

I have reviewed the resubmission and the manuscript has been improved significantly, in terms of the standard of scientific writing and appropriate use of English grammar.